# Two-Layer Heat-Resistant Protective Coatings for Turbine Engine Blades

**Leszek Ułanowicz \* and Andrzej Dudziński**

Air Force Institute of Technology, Księcia Bolesława Street 6, 01-494 Warsaw, Poland
\* Correspondence: leszek.ulanowicz@itwl.pl; Tel.: +48-604-453-280

**Abstract:** One of the most important factors for increasing the durability of turbine engines is the use of turbine blades characterized by the best possible convergence of the thermophysical properties of the protective coating and the base material of the blade. The aim of the research was to evaluate the heat resistance of prototype two-layer protective coatings applied to turbine blades. The inner layer of the coating enables shaping the thermophysical convergence of the coating and the base material of the blade. The outer layer is used for thermal protection of the blade material. The inner layer was applied to the blade by plasma spraying, and the outer layer was diffusion aluminized for the first type by a non-contact gas method, for the second type by a slurry method, and for the third type, the ceramics were plasma sprayed. Turbine blades with prototype coatings were subjected to an engine test, and after the test, macro- and microstructure tests were performed. The tests showed that the prototype protective coating with an inner layer of the MCrAlY type applied to the blade by plasma spraying and an outer layer aluminized by diffusion by a non-contact gas method protects the blade material against oxidation and ensures its thermal insulation.

**Keywords:** aviation; turbine engine; turbine blade; protective coating; heat resistance





## 1. Introduction

The maximum temperature of the gas stream in front of the turbine of a modern gas turbine engine reaches the value of 1473–1623 K. One of the engine turbine parts most exposed to this temperature are its blades. The introduction of nickel-based superalloys into the production of turbine blades, the directional and single crystallization of alloys, and the introduction of blade cooling made it possible to obtain a turbine blade operating temperature of about 1373 K. Work on heat-resistant alloys for turbine blades did not bring the desired effect in the form of an alloy combining the high mechanical properties with satisfactory resistance to high-temperature corrosion [1–3]. The improvement of the mechanical properties of the base alloys causes the deterioration of their environmental resistance [4–6]. Therefore, in order to achieve the required thermo-mechanical properties of the blades, their blades are covered with protective coatings. The diffusion aluminide coatings modified with elements such as chromium, platinum, yttrium, palladium, and hafnium have been used in the greatest applications [7–10]. They have been produced for many years using three methods: batch cementation, Snecma Vapor Phase Aluminizing (SVPA) [3], and Chemical Vapor Deposition (CVD) [9,10]. The use of traditional aluminide coatings for the protection of turbine blades in thermally strained engines turned out to be insufficient because the boundary between the β-NiAl layer and the heat-resistant alloy of the blade core began to melt at a temperature above 1393 K [11,12]. One of the main problems encountered during the operation of aluminide coatings is their thermomechanical fatigue [13,14]. Cyclic deformations caused by temperature gradients in the blades most often lead to fatigue cracks in the shell [15,16].

The protective coatings used as present slow down the process of oxidation and depletion of the monocrystalline component of the substrate from alloying elements. However,

they do not protect the blade walls from overheating because they do not have good insulating properties [17,18]. In operation, the structure of the protective coating and the blade material are often degraded. Examinations of the blades from exploitation show that they contain: oxidation of the protective coating and its diffusion interaction with the native metal, dissolution and globular coagulation of the γ' phase in the nickel alloy, degradation of the blade wall structure, and an intensive decrease in aluminum content, which is responsible for heat resistance [19,20]. The above phenomena cause the formation and propagation of thermal fatigue cracks in the blade. This causes the engines to be taken out of service prematurely. This significantly increases the cost of operating the aircraft and reduces the safety of flights. The analyzes carried out by the authors of this article show that the structure and thermophysical properties of the protective coating must be adapted to the given alloy from which the blade is made [9,21–23]. Therefore, efforts should be made to achieve the best possible convergence of the structure and thermophysical properties of the protective coating and the base material of the blade. In recent years, work has been undertaken on the production of protective coatings with a gradient structure [9,24]. The basic concept of producing layers and functional gradient structures is to combine several materials in a multi-layer system so that their properties exceed those of single-component materials. While maintaining the gradient principle, the functionality of the coatings is responsible for environmental countermeasures (erosion, high-temperature corrosion, loss of adhesion, and the formation of cracks perpendicular to the blade material).

The aim of the work was to design heat-resistant protective coatings for turbine blades, and to assess their condition after an engine test under dynamometer conditions. The scope of the work included making protective coatings for turbine blades made of nickel superalloy, testing the blades on a running engine under dynamometer conditions, testing the microstructure of protective coatings after their engine test, and assessing the quality of the protective coatings obtained. The designed protective coatings on the blade of the turbine blade had a two-layer structure. The first, inner layer was an adhesive layer of the NiCoCrAlY type. NiCoCrAlY overlay coatings, unlike diffusion coatings, provide greater independence from the substrate alloy, but also greater flexibility in design because the compositions can be modified depending on the expected degradation mechanisms. Typical bond coatings contain at least four basic elements plus yttrium (Y), hafnium (Hf), and tantalum (Ta). The presence of a significant amount of Cr gives these coatings excellent corrosion resistance combined with good oxidation resistance. NiCoCrAlY coatings typically exhibit a two-phase β + γ microstructure. The presence of γ increases the ductility of the coating, thereby improving the resistance to thermal fatigue. Ni and Co added together results in better corrosion resistance. Cr provides resistance to hot corrosion, but the amount that can be added is limited by the effect it should have on the substrate and the formation of Cr-rich phases in the coating. The Al content is usually about 10%–12% by volume. NiCoCrAlY also usually contains 1% vol. by weight yttrium (Y), which improves the adhesion of the oxide layer, combines with sulfur, and prevents it from segregating into the oxide layer, which would otherwise damage its adhesion. Hafnium (Hf) additions play a similar role to that of yttrium. The addition of rhenium (Re) in NiCoCrAlY improves the isothermal or cyclic oxidation resistance and thermal cycle fatigue, and tantalum (Ta) increases the oxidation resistance. Due to the presence of a significant amount of Cr in it, it protects the base material of the blade against oxidation and high-temperature corrosion. The advantages of the NiCoCrAlY layer are that it is possible to shape the content of aluminum and chromium and other elements in it and that its properties do not depend on the chemical composition of the blade material [22–26]. This layer is characterized by a low transition temperature from the brittle state to the plastic state, amounting to 473–573 K. The second layer, the outer one, was a diffusion thermal layer of the NiAl type with low thermal conductivity. Heat resistance tests carried out by Kukla et al. [27,28] showed that the oxidized aluminum coating is characterized by excellent durability and tightness of the protective scale, as no scale chips were observed on the surface of the tested coating. They found that the phase structure of the scale consisted mainly of NiAl and $NiAl_2O_4$

intermetallic phases and a stable $\alpha$-$Al_2O_3$ oxide, which improves the corrosion resistance of nickel alloys. They found a significant improvement in the strength response of the MAR 247 nickel alloy by about 200 MPa during cyclic loading at 1173 K. They confirmed that the use of an aluminide layer can effectively protect the parent material against such processes as oxidation, corrosion, and wear. NiAl aluminide has a melting point of 1956 K [13,26]. The outer layer is intended for thermal protection against the high-temperature stream of aggressive fuel combustion products. In the designed coatings, the NiCoCrAlY protective layer was applied to the blade by plasma spraying (APS), and the diffusion aluminization of the protective coating (NiAl) was carried out using the non-contact gas (VPA) and slurry (SLURRY) methods. In the case of the ceramic coating, a ceramic layer was deposited on the NiCoCrAlY layer by plasma spraying (APS). In order for the ceramic coating to have a chance not to spatter during the first thermal cycle, it is important that its thermal expansion is close to the thermal expansion of the substrate. For a coating to be useful, it must also have very low thermal conductivity. Zirconium oxide ($ZrO_2$) stabilized with yttrium oxide ($Y_2O_3$) is widely used for this purpose. The addition of 5%–15% yttrium oxide stabilizes the zirconia in its high-temperature crystalline form, thus avoiding a phase transition in the operating temperature range. Zirconium oxide ($ZrO_2$) ceramic meets both the requirements, with a coefficient of thermal expansion of $11$–$13 \times 10^{-6}$ $K^{-1}$ and a thermal conductivity of approximately 2,3 W/(mK) at 1000 °C for a fully compacted material, and it can be additionally reduced by introducing porosity.

A new element of the research was the fact that blades with protective coatings were tested on a running engine under dynamometer conditions. Previous research practices consisted in annealing samples with protective coatings in an electric furnace.

## 2. Materials and Methods

### 2.1. Materials

The object of the tests were protective coatings applied to the turbine blades of the RD-33 aviation turbine engine made of nickel superalloy, the chemical composition of which is shown in Table 1. These blades are produced in a vacuum foundry furnace using directional crystallization technology using crystalline nuclei (starting cones) to give solidifying directionally columnar grains of the correct crystallographic orientation <001> coinciding with the z axis of the blade [29].

**Table 1.** Chemical composition of turbine blade material–nickel superalloy [29].

| Element Content | Chemical Composition,% by Weight | | | | | | | | | | | | |
|:---:|:---:|:---:|:---:|:---:|:---:|:---:|:---:|:---:|:---:|:---:|:---:|:---:|:---:|
| | **Ni** | **Cr** | **Al** | **Ti** | **Mo** | **W** | **Re** | **Co** | **Nb** | **C** | **Ta** | **Si** | **Mn** |
| Min. | warp | 4.5 | 5.7 | 0.8 | 0.9 | 8.1 | 3.6 | 9.0 | 1.4 | 0.12 | 3.7 | - | - |
| Max | warp | 5.3 | 6.2 | 1.2 | 1.3 | 8.9 | 4.3 | 9.5 | 1.8 | 0.17 | 4.4 | 0.2 | 0.3 |

Amdry 386-4 coating material (Oerlikon Metco Europe GmbH Sp. z o. o. Branch in Poland, Poznań, Poland) was used to make the inner layer of the protective coating of the turbine blade, which was applied to the blades by plasma spraying (coating type NiCoCrAlYHfSi), hereinafter referred to as APS. The composition of the Amdry 386-4 coating material is shown in Table 2. The first type of outer layer is a diffusion aluminized layer produced by the non-contact gas method (VPA), the second type was produced by a slurry method (SLURRY), and for the third one, ceramic powder METCO 204 BN-S was plasma sprayed (zirconium oxide ($ZrO_2$), yttrium oxide ($Y_2O_3$)). $ZrO_2$-$8Y_2O_3$ ceramic powder (Metco 204 BN-S, Oerlikon Balzers Coating Poland Sp. z o. o. Kedzierzyn-Kozle, Poland) had a chemical composition of $ZrO_2$+ 8% by weight. $Y_2O_3$ had a granulation range of $-125 + 11$ μm. Due to the large size of the powder particles, its dimensions were reduced with an average particle size of $d_{V50} = 4.5$ μm. To reduce the size of powder particles, the ball milling process was used, where a single cartridge consisted of: (i) pow-

der, (ii) zirconia balls, (iii) a dispersant preventing the formation of agglomerates, and (iv) ethanol as a cooling medium. The maximum thickness of the protective coating applied to the turbine blade was 160 μm (APS). In the case of a two-layer coating, the thickness of the NiCoCrAlYHfSi (APS) layer was 90–120 μm and that of the NiAl layer was 40–70 μm.

**Table 2.** Chemical composition of Amdry 386-4 coating material.

| Chemical Composition of Amdry 386-4 Coating Material, % by Weight | | | | | | |
|---|---|---|---|---|---|---|
| **Ni** | **Cr** | **Co** | **Al** | **Hf** | **Y** | **Si** |
| 47.8 | 16.4 | 21.9 | 12.1 | 0.5 | 0.6 | 0.4 |

The process of producing coatings by APS plasma spraying was carried out using a Sulzer Metco MultiCoat device equipped with an F4-MB gun. The technological parameters of the NiCoCrAlYHfSi coating process are as follows: the distance between the gun nozzle and the blade: 145 mm; hydrogen flow: 14 dm³/min; argon flow: 65 dm³/min. After depositing the protective coating on the blade, it was subjected to heat treatment and shot peening. This allowed the layers of the coating to diffuse to each other, and thus improved its adhesion. The heat treatment of the coating was carried out in a vacuum oven for 4 h at a heating rate of 10 °C/min at a temperature of 1050 °C. Heat treatment was carried out in a protective atmosphere of argon. The shot peening process was performed with glass balls with a granulation of 230 mesh and a pressure of 4 bar. The protective coating called APS + VPA, APS + SLURRY, APS + TBC was made in two stages. In the first step, the coating called APS, which is described above, was made. In the second stage, in the case of the coating called APS + VPA, diffusion aluminization was performed using the VPA method with the use of CrAl 70/30 coating material and AlF3 activator. The aluminizing process was carried out for 2 h at a heating rate of 7 °C/min at a temperature of 1050 °C. In the case of the APS + SLURRY coating, diffusion aluminization was performed using the suspension method using the Ceral 10 coating material. Drying of the coating layers was carried out in the following cycles: 1 h at room temperature; 15 min at 80 °C; 30 min at 200 °C. In the case of the coating called APS + TBC, METCO 204 BN-S ceramic powder was applied using the APS method.

*2.2. Methods*

Turbine blades with protective coatings applied to them were built into the turbine of the RD-33 turbine engine of a military aircraft. The engine turbine is equipped with 6 blades with a protective coating of each of the four types, i.e., APS, APS + VPA, APS + SLURRY, and APS + TBC. In order to recreate all the loads acting on the turbine blades, an engine test was carried out in dynamometer conditions for 52 h, according to the profile shown in Figure 1.

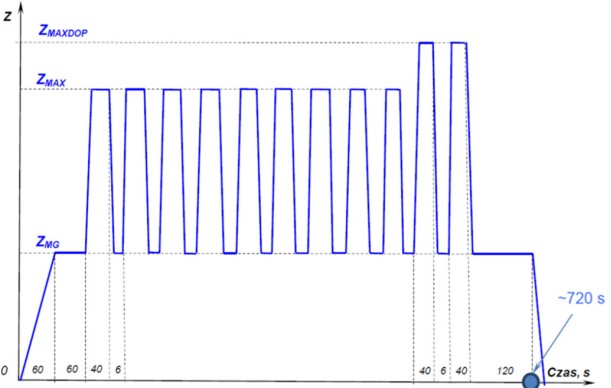

**Figure 1.** The realized profile of the turbojet engine research cycle. Drawing description: Z–operation range; $Z_{MG}$–operation at the idle speed range (low gas); $Z_{MAX}$–operation at the maximum speed range; $Z_{MAX\,DOP}$–operation at the maximum afterburning range [18].

Operation in the ZMG idling speed range took place at a temperature in the range of 700–750 °C. Operation in the ZMAX maximum rotational speed range took place at a temperature in the range of 1100–1200 °C. Operation in the range of maximum afterburning ZMAX DOP took place at a temperature above 1350 °C. The test cycle was repeated a total of 250 times. Individual blades of the rotating turbine were subjected to the same dynamic and thermal loads, which were variable. During one test cycle, the flame of an additional "fire path" (ignition of the fuel in the afterburner assembly) affected the blades with protective coatings for 12 s. This caused an additional thermal load on the protective coatings on the blades. After a 52 h engine test under dynamometer conditions, the blades with protective coatings were subjected to macroscopic and metallographic examinations of the area of the blade's leading edge, and the coating hardness was measured.

Macrophotographs of the protective coatings taken using a Canon EOS 700D camera equipped with a Canon Macro Lens EF 100 mm 1:2.8 L IS USM lens were used for macroscopic examination of the protective coatings.

Prior to metallographic testing of blades with protective coatings, metallographic microsections were made on them. For this purpose, each blade was cut using a metallographic cutter according to the scheme shown in Figure 2.

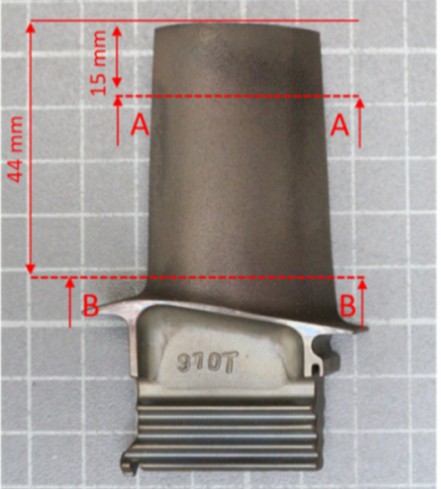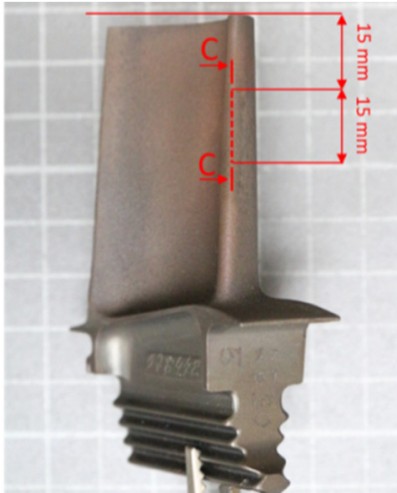

**Figure 2.** Scheme of cutting out fragments of the blade with a prototype protective coating for the execution of metallographic specimens of the blade material.

The microstructure of the protective coating was tested using a scanning electron microscope (type Quanta 3D FEG (SEM/FIB, FEI Company, Hillsboro, OR, USA) equipped with the EDS system (FEI Company, Hillsboro, OR, USA) and software SPIP (Image Metrology, Kongens Lyngby, Denmark), enabling measurements of the thickness of the coatings and their component layers, and using a metallographic microscope. Microhardness tests of protective coatings were performed using the Future-Tech FM-700 microhardness tester (Future-Tech Corp., Tokyo, Japan) using the Vickers HV 0.01 method.

The thermal diffusivity of the protective coatings was determined using the pulsed laser method using a low-temperature diffuser LFA 467 and a high-temperature diffuser LFA 427 by Netzsch. The measurement was made of the samples cut from a blade with a side length of 10 mm and a thickness of about 1 mm (Figure 3). The temperature range of the measurement was from 25 to 1200 °C. The measurement was made at 100 °C in an argon atmosphere. The measurement was performed at a pulse voltage of 450 V and a pulse duration of 0.6 ms. At each temperature, 3 shots were taken with 2 min intervals.

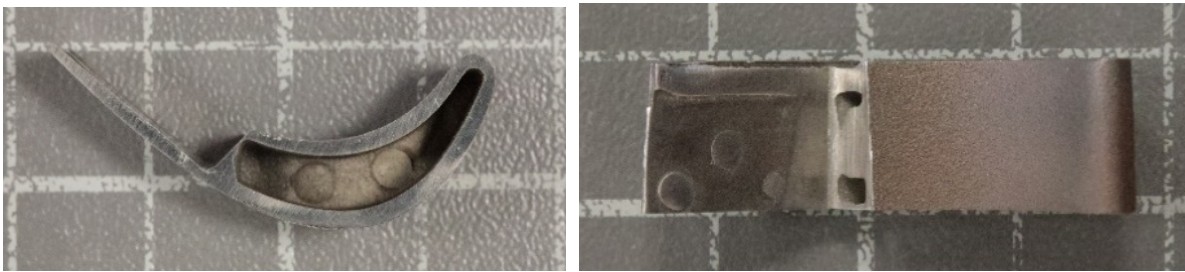

**Figure 3.** A fragment of a blade with a sample cut out for testing the diffusivity of a protective coating.

### 3. Results

*3.1. Initial Structure of Protective Coatings after Their Application to Turbine Blades*

An exemplary microstructure of the protective coating produced by plasma spraying called APS after its heat and mechanical treatment (shot peening) is shown in Figure 4a, while the protective coating called APS + VPA is in Figure 4b, the protective coating called APS + SLURRY is in Figure 4c, and the protective coating called APS + TBC is in Figure 4d.

**Figure 4.** An example of the microstructures of the protective coatings called (**a**) APS, (**b**) APS+VPA, (**c**) APS + SLURRY, (**d**) and APS + TBC after their heat and mechanical treatment.

In Figure 4a–c, the porosity of the outer layer of the resulting protective coating is noticeable. The porosity of the outer layer of the protective coating is probably due to the fact that the coating material was not uniform in size and had an irregular shape with a large number of sharp edges. This is a consequence of their crushing, and then grinding in ball mills. This form of the starting material may, to some extent, reduce the efficiency of the powder dosing process in the plasma stream, which has a significant impact on the properties of the coating, e.g., porosity and tightness. Additionally, the presence of a discontinuity zone in the areas of the coating affects its porosity. It is estimated that the resulting coating has a porosity of up to 15%. Of course, this is an imperfection of the obtained structure, thus affecting its durability. However, on the outer layer, a network of cracks was not observed, which would cause delamination of the coating. It should be mentioned here that these coatings were made by experimental production (prototypes). In order to eliminate this type of imperfection of protective coatings, further technological research should be carried out to optimize the technical conditions for applying coatings, thus improving their adhesion, using numerical models and modern laboratory research methods.

### 3.2. Thermal Diffusivity of the Blade Material and Protective Coatings

The results of thermal diffusivity tests of the blade material (nickel superalloy) of the engine turbine and protective coatings are shown in Figure 5.

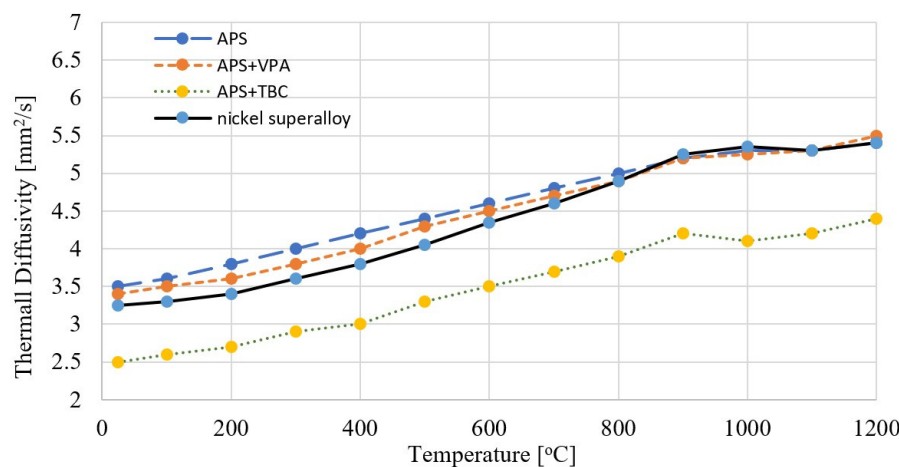

**Figure 5.** Thermal diffusivity of the blade material (nickel superalloy) of the engine turbine and protective coatings.

The diffusivity of the blade material (nickel superalloy) in terms of temperature ranges from 3.258 to 5.45 mm$^2$/s. In the temperature range up to 1200 °C, the turbine blade material does not show clear signs of phase transformations. The diffusivity of the APS type protective coating in terms of temperature ranges from 3.542 to 5.412 mm$^2$/s. The diffusivity of the APS + VPA protective coating in terms of temperature ranges from 3.429 to 5.495 mm$^2$/s. The diffusivity of the APS + SLURRY protective coating in terms of temperature ranges from 3.844 to 6.614 mm$^2$/s. The diffusivity of the APS + TBC protective coating in terms of temperature ranges from 2.523 to 4.416 mm$^2$/s.

### 3.3. Results of Macroscopic Tests of Protective Coatings after Engine Test in Dynamometer Conditions

Protective coatings made using the APS method had, on the leading edge of the blade, at a distance of about 10 mm from the tip of the blade along a length of about 15 mm (highest temperature impact zone), single spot damages (three blades) and local flaking and shallow coating defects (the remaining blades). A representative image of the protective coating on the leading edge of the blade is shown in macrophotographs in Figure 6.

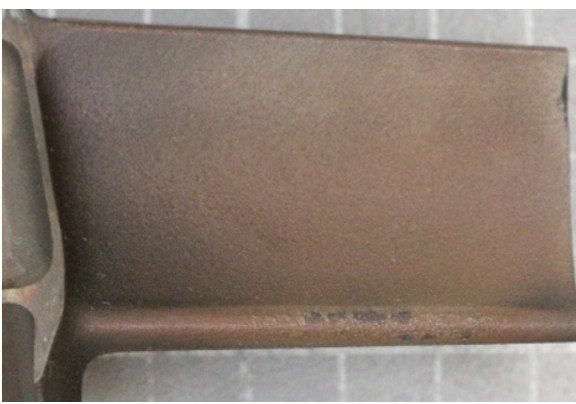
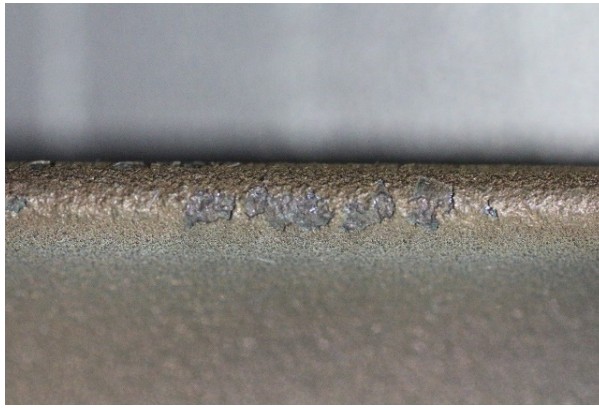

**Figure 6.** Macrophotograph of the leading edge of one of the blades with a protective coating made using the APS method.

Protective coatings made using the APS + VPA method had no visible damage, peeling, defects, or discoloration along the entire length of the leading edge of the blade. A representative image of the protective coating on the leading edge of the blade is shown in the macrophotograph in Figure 7.

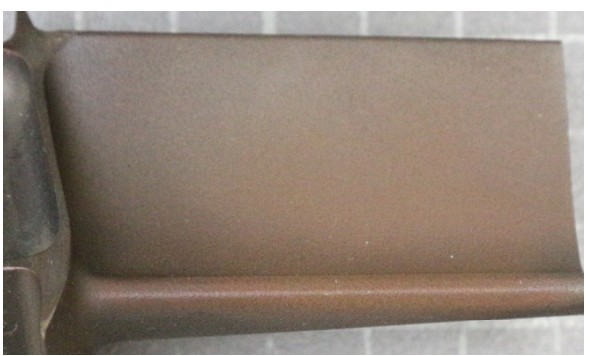
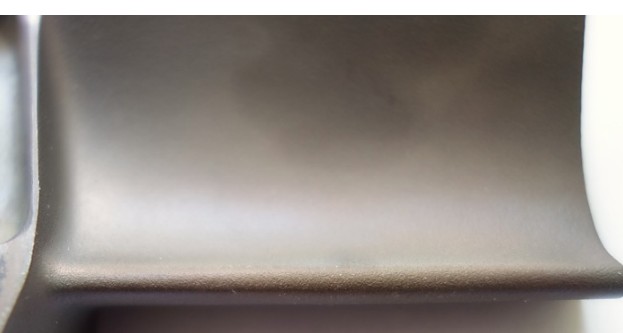

**Figure 7.** Macrophotographs of the leading edge of blades with a protective coating made by the APS + VPA method.

The protective coatings made with the APS + SLURRY method had, on the leading edge of the blade, at a distance of about 10 mm from the tip of the blade at a length of about 15 mm (highest temperature impact zone), damage ranging from local flaking (one blade) to extensive and deep coating defects, with areas exposing the base alloy (the remaining blades). Representative images of the leading edge of blades with a protective coating made using the APS + SLURRY method are shown in the macrophotograph in Figure 8. In the macrophotograph in Figure 8, we can see local flaking and extensive, deep defects of the protective coating on and around the leading edge.

The protective coatings made with the APS + TBC method had, on the leading edge of the blade, damage ranging from local flaking of the ceramic coating (four blades) to extensive losses of the ceramic coating along the entire length of the leading edge (two blades). Representative images of the leading edge of the blades with a protective coating made by the APS + TBC method are shown in the macrophotographs in Figure 9.

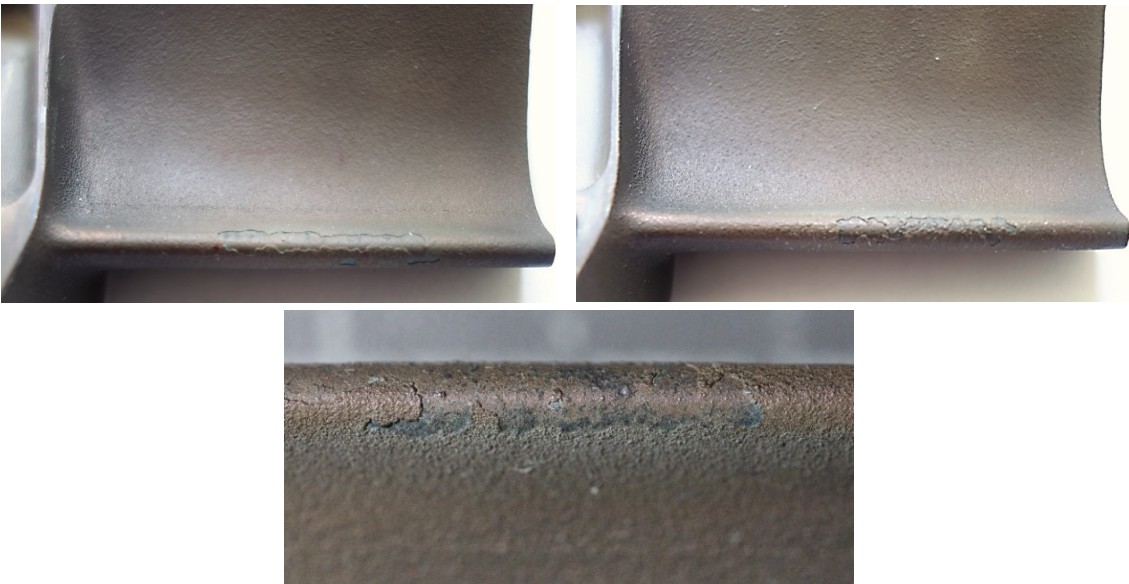

**Figure 8.** Macrophotographs of the leading edge of the blade with a protective coating made using the APS + SLURRY method.

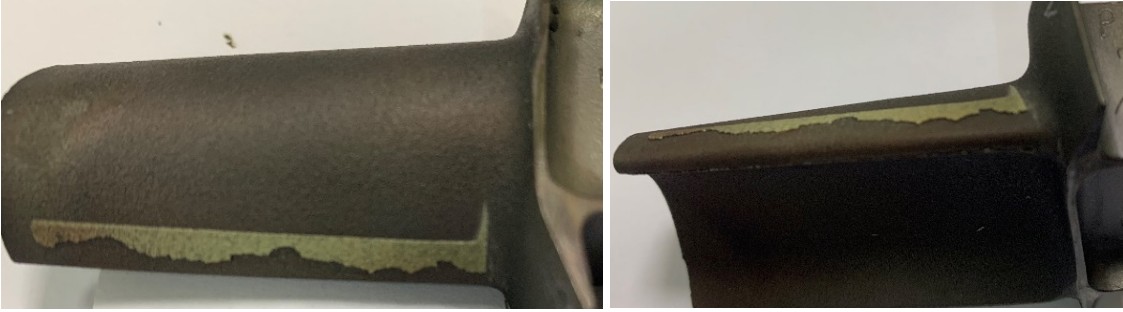

**Figure 9.** Macrophotographs of the leading edge of the blade with a protective coating made by the APS + TBC method.

### 3.4. Microhardness of the Protective Coating on the Leading Edge of the Blade after the Engine Test under Dynamometer Conditions

Table 3 presents the results of measurements of the thickness of protective coatings and their hardness HV0.01 in the leading edge zone, i.e., in the area of the blade most exposed to the impact of the exhaust gases.

Figure 10 shows a graph showing the distribution of HV0.01 microhardness on one of the protective coatings made using the APS method. The average value of microhardness HV0.01 of this protective coating in the longitudinal section of the blade (see C–C in Figure 2) is 509 (SD deviation 50). Figure 11 shows a graph showing the distribution of HV0.01 microhardness on one of the protective coatings made using the APS+VPA method. The average value of microhardness HV0.01 of this protective coating in the longitudinal section of the blade (see C–C in Figure 2) is 595 (SD deviation 68). Figure 12 shows a graph showing the distribution of HV0.01 microhardness on one of the protective coatings made using the APS+ SLURRY method. The average value of microhardness HV0.01 of this protective coating in the longitudinal section of the blade (see C–C in Figure 2) is 414 (SD deviation 42). Figure 13 shows a graph showing the distribution of HV0.01 microhardness on one of the protective coatings made using the APS+ SLURRY method. The average value of microhardness HV0.01 of this protective coating in the longitudinal section of the blade (see C–C in Figure 2) is 492 (SD deviation 77).

**Table 3.** The results of measurements of the thickness of protective coatings and their hardness HV0.01 in the leading edge zone.

| Coating Type | No. of the Blade in the Turbine Palisade | Coating Thickness-SEM Measurements (μm) | | Hardness HV0.01 | Medium Hardness HV0.01 |
|---|---|---|---|---|---|
| | | Cross-Section | Longitudinal Section | | |
| APS | 11 | 44 | 62 | 509 | 478 |
| | 37 | 75 | 70 | 494 | |
| | 65 | 73 | 66 | 483 | |
| | 71 | 73 | 76 | 503 | |
| | 76 | 87 | 78 | 385 | |
| | 83 | 98 | 90 | 492 | |
| APS + VPA | 3 | 144 | 128 | - | 530 |
| | 5 | 132 | 132 | 502 | |
| | 30 | 141 | 138 | 523 | |
| | 32 | 128 | 126 | 543 | |
| | 75 | 134 | 128 | 504 | |
| | 84 | 146 | 138 | 595 | |
| APS + SLURRY | 13 | locally 46–82 | locally 92 | 453 | 424 |
| | 27 | 76 | 108 | 443 | |
| | 55 | locally 35–88 | Locally 131 | 383 | |
| | 57 | 84 | 112 | 402 | |
| | 61 | do 68 | do 74 | 414 | |
| | 79 | locally 94 | locally 122 | 449 | |
| APS + TBC | 15 | 143 | 138 | 492 | 497 |
| | 17 | locally 89–96 | locally 142 | 449 | |
| | 21 | 144 | 148 | 498 | |
| | 7 | 134 | 128 | 510 | |
| | 43 | 128 | 119 | 502 | |
| | 45 | 136 | 124 | 523 | |

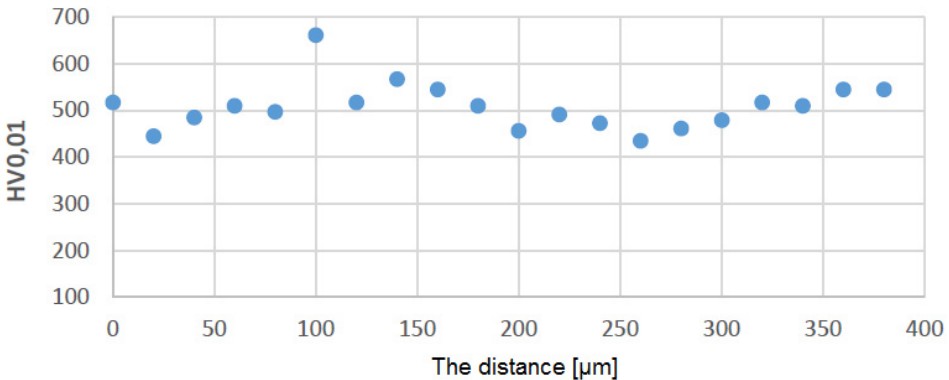

**Figure 10.** Distribution of microhardness on the leading edge for the protective coating made using the APS method.

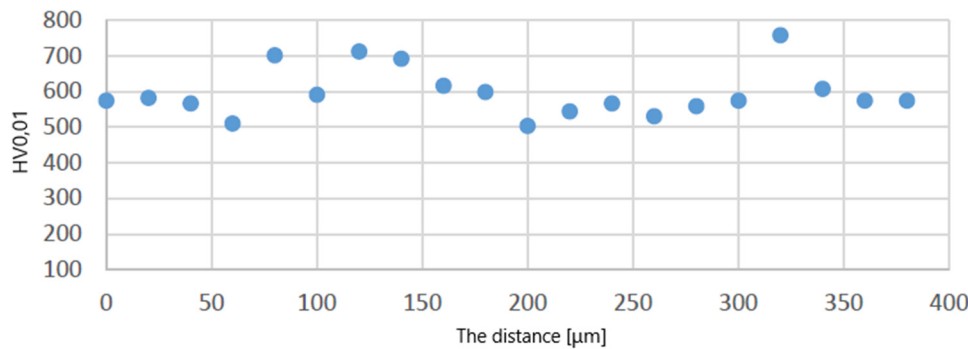

**Figure 11.** Distribution of microhardness on the leading edge for the protective coating made using the APS + VPA method.

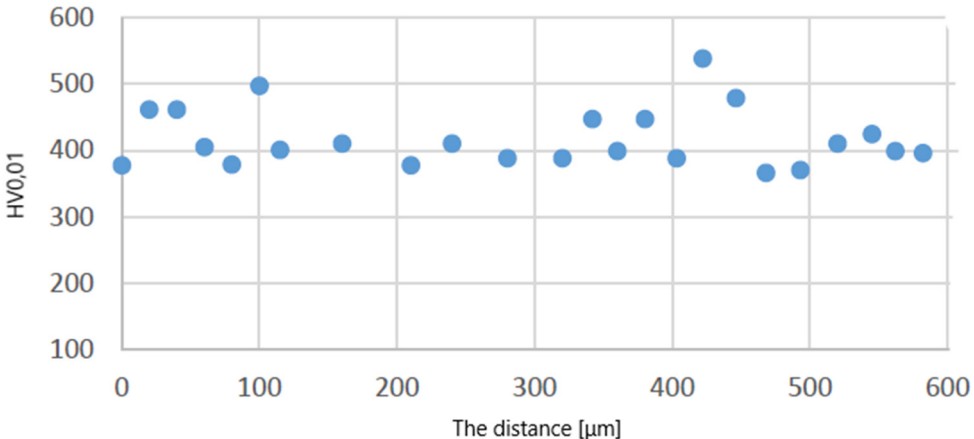

**Figure 12.** Distribution of microhardness on the leading edge for the protective coating made using the APS + SLURRY method.

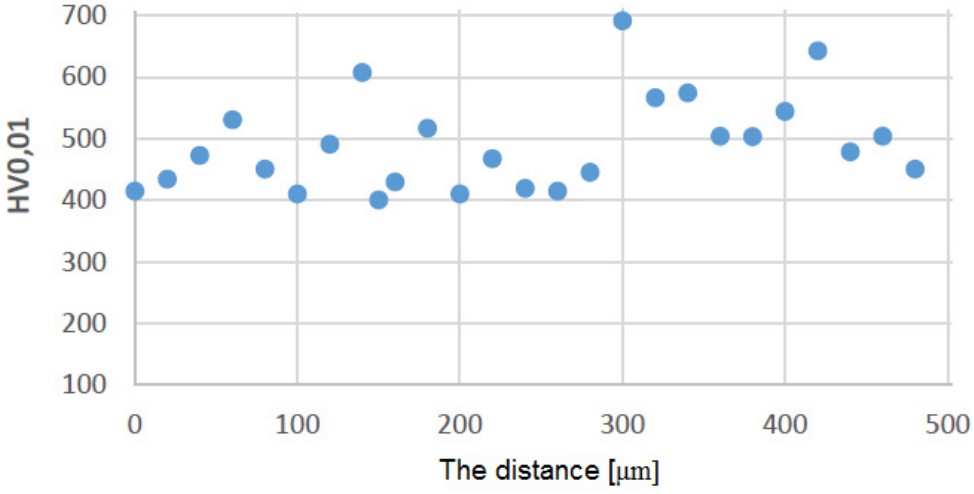

**Figure 13.** Distribution of microhardness on the leading edge for the protective coating made using the APS + TBC method.

The distribution of the average hardness HV0.01 of the protective coatings did not show significant differences in the hardness on the leading edge of the turbine blades, so it can be concluded that the base material was not overheated.

### 3.5. The Structure of the Protective Coatings on the Leading Edge of the Blade after the Engine Test under Dynamometer Conditions

Representative structures of the protective coatings on the leading edge of the turbine blade in the cross-section, A–A, of the blade (see Figure 3) made using the APS, APS + SLURRY, APS + VPA, and APS + TBC methods are shown in Figure 14 and in the longitudinal section, C–C, of the blade in Figure 15.

The coating made using the APS method (Figures 14a and 15a) is characterized by a strongly developed surface, varied thickness distribution, and the presence of oxidation products over the entire cross-section of the coating. The coating contains numerous oxides arranged perpendicularly to the surface and discontinuities. The results of the analysis of the content of aluminum Al and chromium Cr in selected areas of the protective coating made using the APS method on the leading edge of the blade indicate that the average distribution of Al in the protective coating at a depth of 5 μm is uniform at a level that does not exceed 8% by weight, with an average Cr content of above 14% by weight. The average distribution of Al in the protective coating at a depth of 30 μm is at a level that does not exceed 10% by weight, with the average Cr content at a level that does not exceed 12% by weight. A significant content of nickel and cobalt, as well as precipitates of silicon and molybdenum, are observed at both depths of the coating. Minor amounts of hafnium and yttrium were found.

The protective coating made using the APS + SLURRY method (Figures 14b and 15b) is characterized by a typical sprayed microstructure with the presence of an outer layer enriched in Al. The coating has a very strongly developed surface, and there are many cracks perpendicular to the surface and reaching the alloy of the substrate. Inside the coating, there are numerous oxides and delaminations. Local corrosion pits under the coating were observed. The results of the analysis of the content of aluminum Al and chromium Cr in selected areas of the protective coating made using the APS + SLURRY method indicate that the average distribution of Al in the protective coating at a depth of 5 μm is about 14% by weight, with the average Cr content being below 8% by weight. The average distribution of Al in the protective coating at a depth of 30 μm is about 17% by weight, with the average content of Cr not exceeding 10% by weight. In both zones of the protective coating, an evenly distributed amount of chromium and significant amounts of nickel and cobalt were observed. Silicon-rich precipitates were also found.

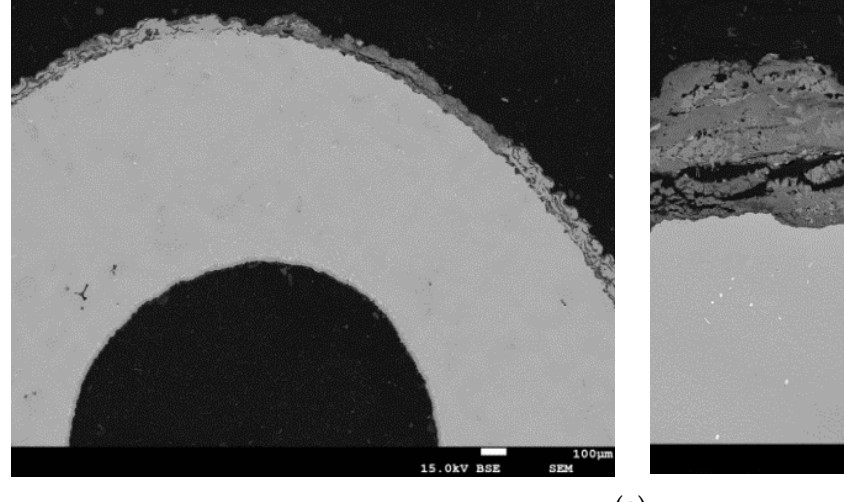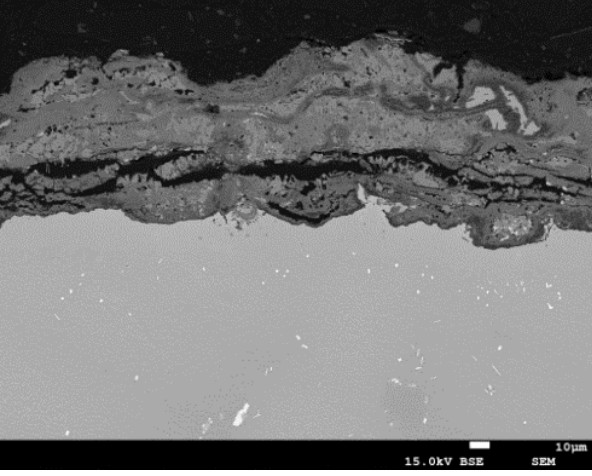

(a)

**Figure 14.** *Cont.*

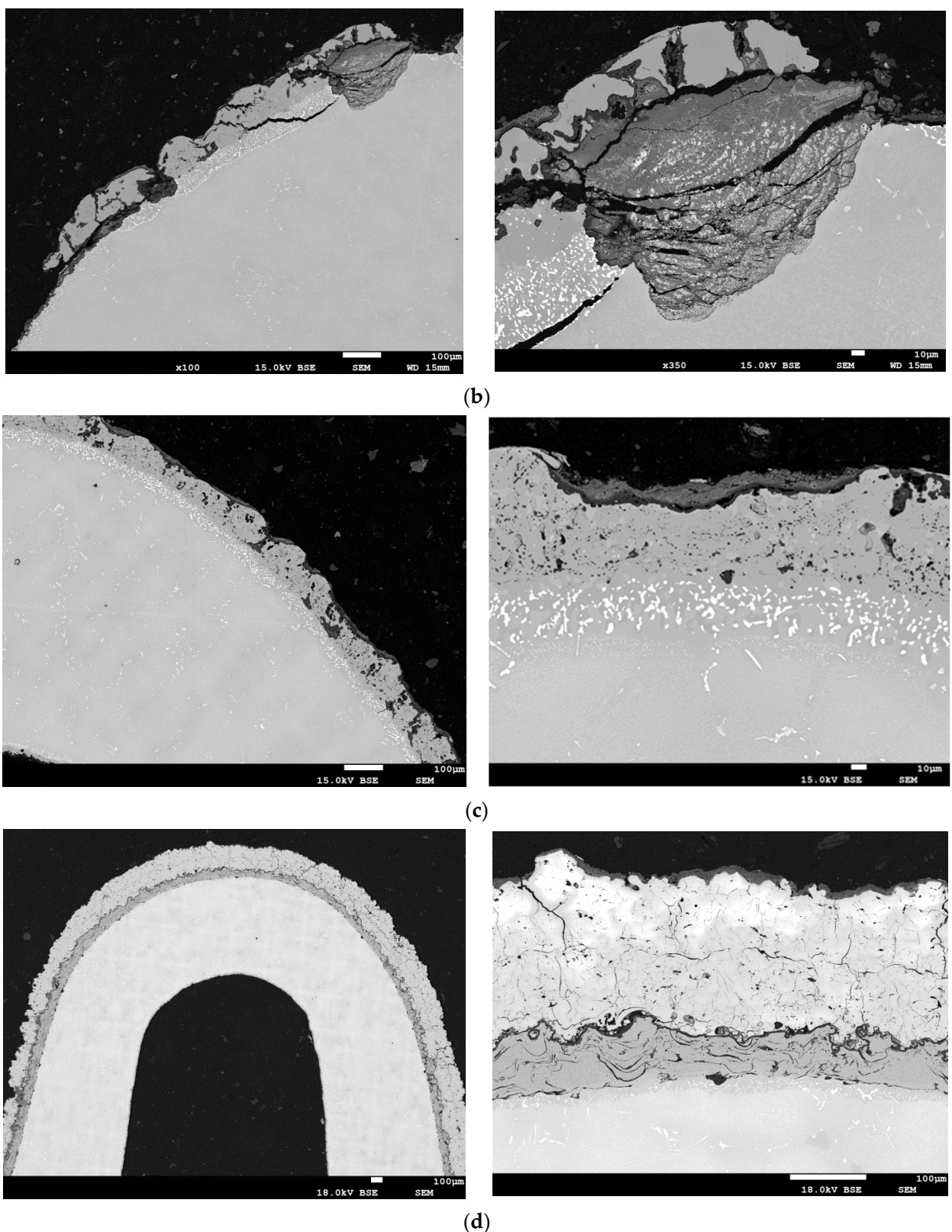

**Figure 14.** The structure of the protective coatings in the blade cross-section A–A (see Figure 2) made by the methods: (**a**) APS; (**b**) APS + SLURRY; (**c**) APS + VPA; (**d**) APS + TBC.

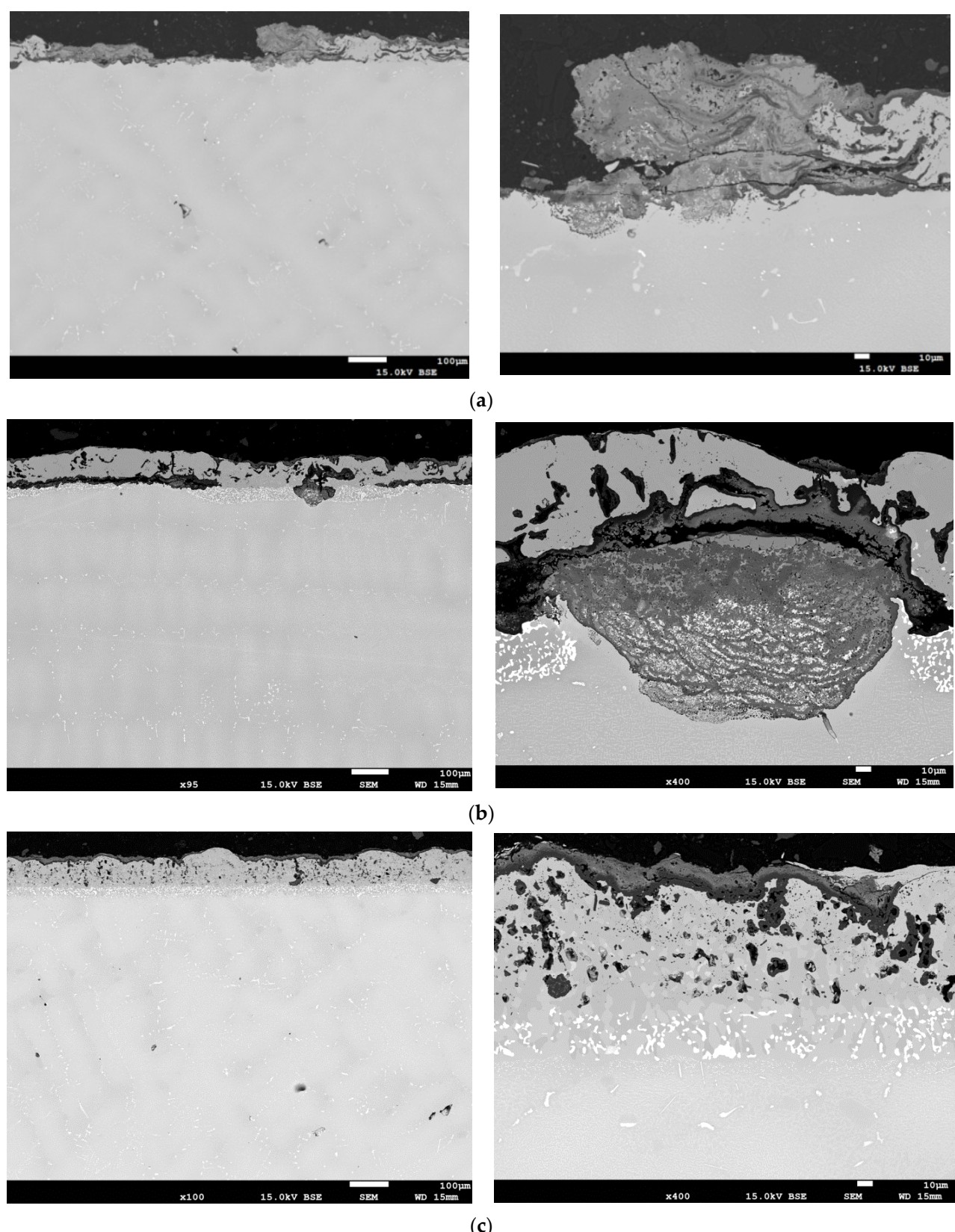

**Figure 15.** *Cont.*

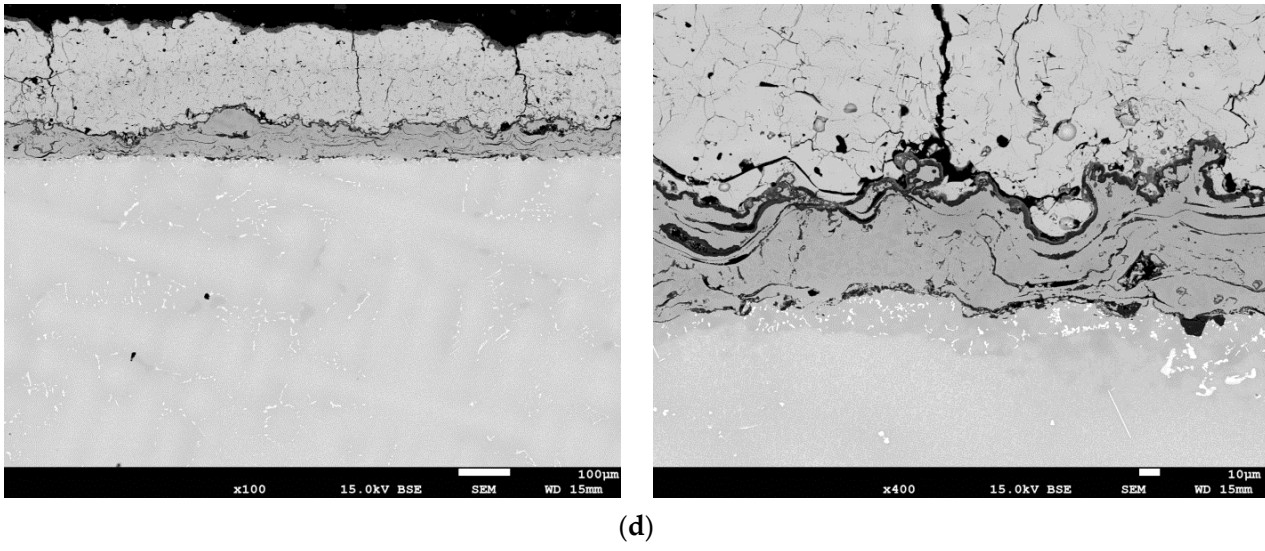

**(d)**

**Figure 15.** The structure of the protective coatings in the longitudinal section of the C–C blade (see Figure 2) made by the methods: (**a**) APS; (**b**) APS + SLURRY; (**c**) APS +VPA; (**d**) APS + TBC.

The protective coating made using the APS + VPA method (Figures 14c and 15c) is characterized by a uniform thickness and a two-phase microstructure. The microscopic examination of the coating showed that it is homogeneous and even. There are oxides and local porosity near the surface. The results of the analysis of the chemical composition in selected areas of the protective coating made using the APS + VPA method on the leading edge of the blade indicate that the average distribution of Al in the protective coating at a depth of 5 µm is about 11.5% by weight, with an average Cr content of about 10.5% by weight. The average distribution of Al in the protective coating at a depth of 30 µm is about 16.5% by weight, with the average Cr content not exceeding 10.5% by weight. In both zones of the protective coating, evenly distributed contents of chromium, nickel, and cobalt were observed. Precipitates containing silicon, vanadium, hafnium, and yttrium were found.

A protective coating made using the APS + TBC method (Figures 14d and 15d) is characterized by the presence of an interlayer adhering to the substrate with a microstructure characteristic of sprayed coatings. A thin layer of TGO was observed on the border with a well-adhering ceramic coating, as well as deposits and cracks on the surface of the ceramic coating. Localized corrosion from the surface to the transition zone was observed. The coating contained layered oxides and discontinuities and porosities around them. The results of the analysis of the chemical composition of the protective coating made by the APS + TBC spraying method on the leading edge of the blade indicate that the average distribution of Al in the protective coating at a depth of 30 µm is about 13% by weight, with an average Cr content of about 14% by weight. In both zones of the protective coating, evenly distributed and significant contents of aluminium, zirconium, chromium, nickel, and cobalt was observed. Precipitates containing silicon and yttrium were also found.

## 4. Discussion

Protective coatings of the APS type after the engine test under dynamometer conditions show flaking on the leading edges of the blades. The microstructure of the coatings is homogeneous and even, which is characteristic of thermally sprayed coatings. Corrosion products of APS coatings extend to the border with the substrate. The analysis of the chemical composition of the coating shows that the distribution of Al in the coating is uniform at a level below 8% by weight, with a Cr content of 9%–11% by weight. The outer zone of the shell is slightly poorer in aluminum in relation to that of the zone closer to the base material of the blade. In the zone closer to the native material of the blade, small contents of nickel and cobalt, as well as precipitates of silicon, hafnium, and yttrium, are

observed. The APS-type protective coating has significantly depleted the elements Al, Cr, and Ni. The microstructure of the native material of the blade on its leading edge after the engine test is characterized by the loss of the hexagonal form and the acquisition of a spherical shape (coagulation) by particles of the gamma prime phase (marked γ′) (see Figure 16). The γ′ phase consists of ordered, cubic, plane-centered precipitates of the Ni$_3$Al compound that are embedded in the matrix material. It plays a key role in giving the superalloy strength.

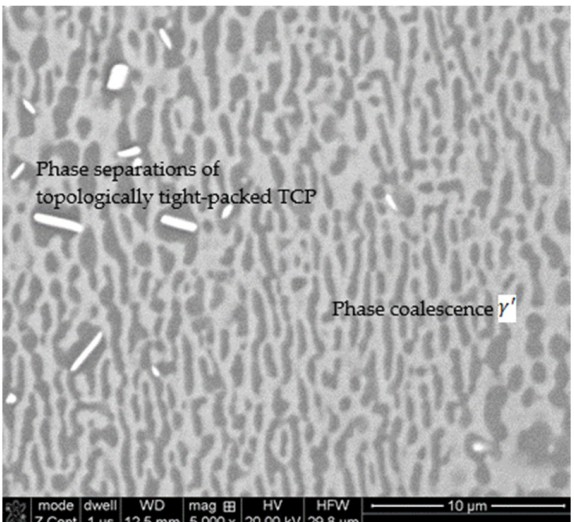 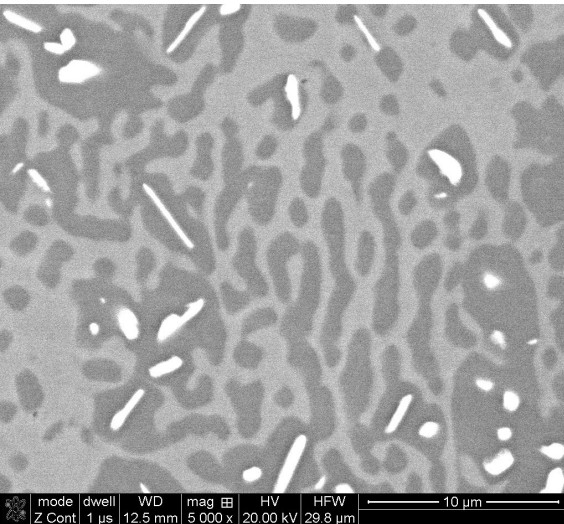

**Figure 16.** The microstructure of the γ/γ′ phases in the parent material on the leading edge of the blade covered with an APS type protective coating.

In the middle of the blade height (left photo), the microstructure has a distinct plate-like structure (the so-called raft structure) resulting from the coalescence of the reinforcing γ′ phase. The raft structure of the γ′ phases are long and oriented perpendicularly to the blade axis. At the tip of the blade, the microstructure (right photo) also shows clear plate-likeness, but the γ′ rafts are thicker and shorter than they are in the center of the blade. Most of the γ′ phase rafts are perpendicular to the blade axis, but in some areas, they are oriented along the blade axis. There is no clearly marked separation of the topologically tight-packed phases and undesirable agglomerates (coalescence), which indicates that the native material of the blades was not overheated. Porosity was not tested.

The protective coatings produced on the blades by the APS + VPA method after the engine test under dynamometer conditions had no damage on the leading edge. The coating is characterized by a uniform thickness and a two-phase microstructure. Microscopic examination of the coating showed that it is homogeneous and even. There are oxides and local porosity near the surface. The analysis of the chemical composition showed that the distribution of Al in the protective coating is uniform at above 15% by weight. In both zones of the protective coating, there is evenly distributed chromium with a content of 12% by weight and significant contents of nickel and cobalt. The protective coating contains Al$_2$O$_3$, Ni$_3$Al, NiAl phases, and NiAl$_2$O$_4$ spinels, i.e., the heat-resistant β-NiAl phase was preserved in the coating. The protective features of the coating were not exhausted during the engine test, and the coating is characterized by very good heat resistance. An example (characteristic) microstructure of the blade material on its leading edge covered with a protective coating of the APS+VPA type is shown in Figure 17.

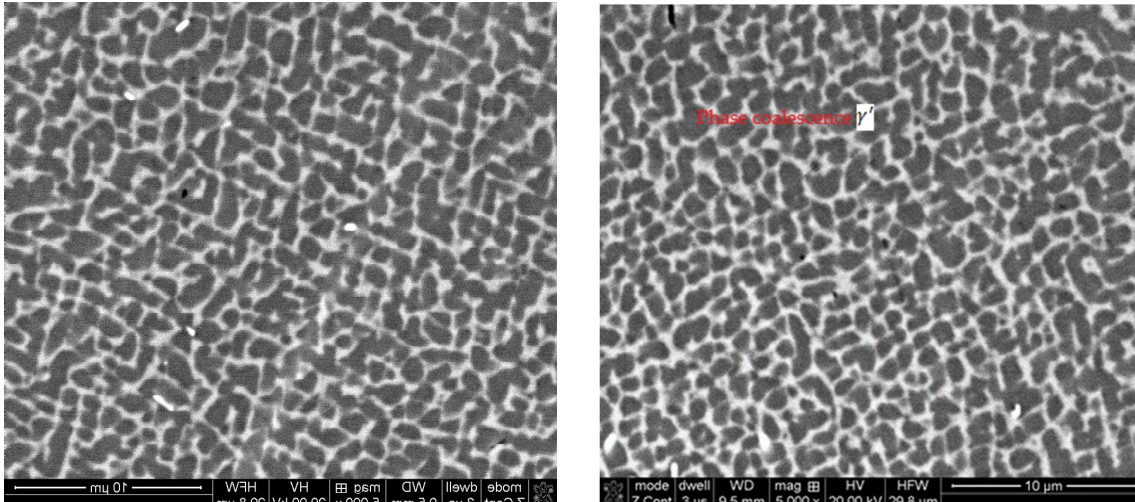

**Figure 17.** The microstructure of the γ/γ′ phases in the parent material on the leading edge of the blade covered with an APS + VPA type protective coating.

The γ/γ′ microstructure (ordered separations of the γ′ strengthening phase in the γ matrix) maintained the cubic morphology with the same periodicity. Slightly coagulated precipitates of the γ′ phase and no effect of dissolving the finely dispersed precipitates of the γ′ phase in the matrix of the γ solid solution prove the lack of thermal degradation of the structure of the native material of the blade. There is no clear shallowness (the so-called raft structure) resulting from the fusion of the reinforcing γ′ phase.

In the case of the protective coating made with the APS + SLURRY method, it can be stated that too much activity of the Al source in the SLURRY aluminizing process leads to a significant increase in the brittleness of the coating at a temperature below 750 °C [12,22], which is visible during its operation due to numerous cracks. Oxidized zones were found in the coatings, as well as areas of corrosion on the border with the transition (diffusion) zone. The analysis of the chemical composition of the coating showed that its outer zone is slightly poorer in aluminum in relation to that of the zone closer to the base material of the blade. The distribution of Al in the outer zone of the coating is at the level of 16%–18% by weight, and in the inner zone it is above 20% by weight. In both zones of the protective coating, there is evenly distributed chromium at the level of 10%–12% and significant contents of nickel and cobalt. There is a residual phase of protective $Al_2O_3$ oxide and $Ni_3Al$ compound in the protective coating. An example of the microstructure of the blade material on its leading edge covered with a protective coating of the APS + SLURRY type is shown in Figure 18.

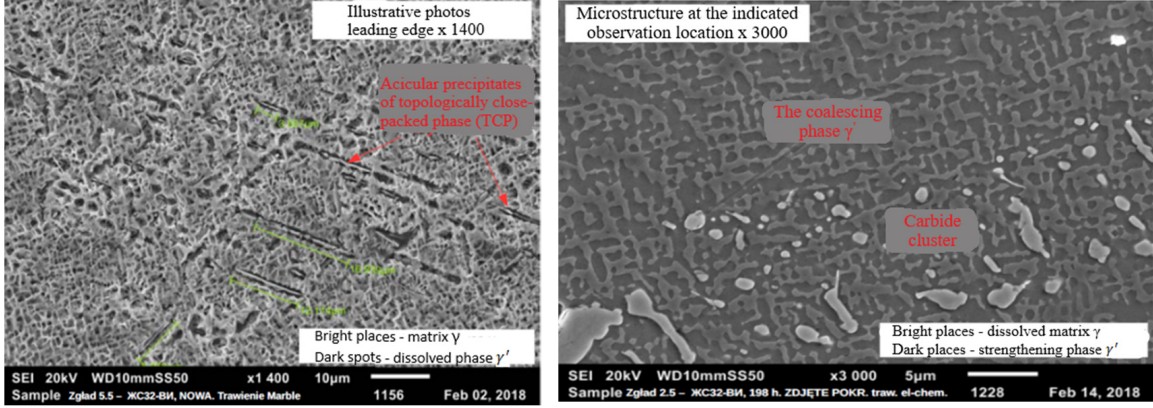

**Figure 18.** The microstructure of the blade material on its leading edge covered with a protective coating of the APS + SLURRY type.

The microstructure of the base material of the blade is characterized by unacceptable separations of the topologically tight-packed phase and undesirable agglomerates (coalescences), indicating local overheating. Porosity was not tested. The coatings on the blades produced by the APS + TBC method had no damage on the leading edge throughout the engine test. The ceramic coating has excellent adhesion to the interlayer, and the interlayer itself is adhesive to the substrate. A well-shaped TGO (Thermally Grown Oxide) layer is observed between the interlayer and the ceramic layer. Small oxidation zones and small areas of corrosion were found in the coating at the border of the transition zone. The analysis of the chemical composition showed that the distribution of Al and Cr in the NiCoCrAlYHfSi layer of the coating is uniform, with Al at the level of 11%–13% by weight and Cr at the level of 12%–14% by weight. The coating contains evenly distributed nickel with a significant content of cobalt.

## 5. Conclusions

Based on the engine test carried out under dynamometer conditions of protective coatings applied to turbine blades using the APS, APS + VPA, APS + SLURRY, and APS + TBC methods and their microstructure testing after the engine test, it can be concluded that the protective properties of the coatings made using the APS + VPA and methods APS + TBC were not exhausted. The heat-resistant β-NiAl phase was preserved in both types of protective coatings. Thus, these coatings retained their heat resistance.

The coatings produced by the APS + TBC method have very interesting properties: that the ceramic coating has excellent adhesion to the interlayer, and the interlayer itself is adhesive to the substrate. A well-shaped TGO (Thermally Grown Oxide) layer is observed between the interlayer and the ceramic layer.

The proposed composition of protective coatings made using the APS + VPA and APS + TBC methods enabled us to achieve the required thermo-mechanical properties of the blade, guaranteeing the protection of the blade against oxidation at high operating temperatures. The desired convergence of the thermo-mechanical properties of the protective coating structure and the base material was achieved, i.e., a heat-resistant base alloy with high mechanical properties and a heat-resistant protective coating with high resistance to oxidation and high-temperature corrosion. The composition of protective coatings made using the APS + VPA and APS + TBC methods increases their heat resistance, thermal insulation, and erosion resistance and increases the durability of the turbine blades and the reliability of the turbojet engine.

The original achievement of the work is the production of two-layer protective coatings on the turbine blades of the RD-33 aircraft engine with an internal adhesive layer of the NiCoCrAlYHfSi type and an external aluminide NiAl or thermal $ZrO_2$-$8Y_2O_3$ diffusion layer and their testing in an equivalent engine test under dynamometer conditions.

**Author Contributions:** Conceptualization, L.U. and A.D.; methodology, L.U. and A.D.; validation, L.U. and A.D.; formal analysis, L.U. and A.D.; investigation, L.U.; resources, A.D.; data curation, L.U. and A.D.; writing—original draft preparation, L.U.; writing—review and editing, L.U.; visualization, A.D.; supervision, L.U.; project administration, L.U.; funding acquisition, L.U. and A.D. All authors have read and agreed to the published version of the manuscript.

**Funding:** The work was financed and carried out as part of the project, No. DOB-BIO8/04/01/2016, entitled "Coatings with increased heat resistance on high-pressure turbine blades of RD-33 engines", financed by the Polish National Center for Research and Development.

**Institutional Review Board Statement:** Not applicable.

**Informed Consent Statement:** Not applicable.

**Data Availability Statement:** Data confirming the reported results can be found in the Library of the Air Force Institute of Technology in Warsaw. The data generated during the tests are available in the form of reports in the Library of the Air Force Institute of Technology.

**Conflicts of Interest:** The authors declare no conflict of interest.

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
