# Peer review of "Two-Layer Heat-Resistant Protective Coatings for Turbine Engine Blades"

_coatings, doi:10.3390/coatings13030588_

Round 1
Reviewer 1 Report
The topic of this article is relevant. It is known that heat-resistant materials are used for the manufacture of turbine blades of aircraft engines. But to protect the working surface of the blades from oxidation, protective coatings must be formed on them. Today there is a lot of work in this direction. The approaches to the formation of protective coatings used in this work have the right to life. But I would like to draw the attention of the authors to my comments.
1. Unfortunately, in the literature review, I did not see references to the works of material scientists from Rzeszów Polytechnic (scientific school of professor J. Sieniawski). They have an excellent experimental base, they not only form turbine blades of aircraft engines from high-nickel alloys, but also developed unique technologies for forming protective coatings of a gradient structure. I think it is necessary to compare your approaches to coating with theirs.
2. The goal in the work is set clearly. You have correctly analyzed the results, but the "Conclusions" section repeats the points from the "Discussion" section. I believe that the conclusions should be structured and specified. They should make it clear that you have achieved your goal.
3. In the photo presented in fig. 10, 11, 12, 13, 22 increases are not marked or are not visible. This impairs the informativeness of the pictures.
After corrections and additions, the article can be published.
Author Response
- Unfortunately, in the literature review, I did not see references to the works of material scientists from Rzeszów Polytechnic (scientific school of professor Sieniawski). They have an excellent experimental base, they not only form turbine blades of aircraft engines from high-nickel alloys, but also developed unique technologies for forming protective coatings of a gradient structure. I think it is necessary to compare your approaches to coating with theirs.
The works of materials scientists from the Rzeszów University of Technology (prof. J. Sieniawski's scientific school) were analyzed. The team presented several dozen works in the field of aluminide coatings. They describe research on the production of protective coatings with a gradient structure, especially with the use of palladium and hafnium coatings in the oxidation zone. The bibliographies have been supplemented with the following items:
- Kwolek, P.S.; Kolek, Ł.; Zagula-Yavorska, M.; Morgiel, J.A.; Romanowska, J. Effect of Pd and Hf co-doping of aluminide coatings on pure nickel and CMSX-4 nickel superalloy. Archives of Civil and Mechanical Engineering 2018, Volume 18 (4), pp. 1421-1429.
- Goral, M.; Pytel, M.; Ochal, K.; Drajewicz, M.; Kubaszek, T.; Simka, W.; Nieuzyla, L. Microstructure of Aluminide Coatings Modified by Pt. Pd, Zr and Hf Formed in Low-Activity CVD. Process. Coatings 2021, Volume 11, 421. https://doi.org/10.3390/ coatings11040421.
In the introduction, the following text was added: In recent years, work has been undertaken on the production of protective coatings with a gradient structure [9, 24]. The basic concept of producing layers and functional-gradient structures is to combine several materials in a multi-layer system so that their properties exceed those of single-component materials. While maintaining the gradient principle, the functionality of the coatings is responsible for environmental countermeasures (erosion, high-temperature corrosion, loss of adhesion, formation of cracks perpendicular to the blade material).
- The goal in the work is set clearly. You have correctly analyzed the results, but the "Conclusions" section repeats the points from the "Discussion" section. I believe that the conclusions should be structured and specified. They should make it clear that you have achieved your goal.
Proposals have been redrafted in order to organize them. Removed e.g. sentences:
“In these coatings there is a phase of the protective oxide Al2O3, and also Ni3Al compounds and a mixture of oxides. These coatings retained their heat resistance. It should be added here that the disappearance of the β-NiAl phase from the coating and the reduction of the aluminum content in it below 8% by weight eliminates such a protective coating from further work. In the case of the protective coating made with the APS+SLURRY method, it can be stated that too high activity of the Al source in the SLURRY aluminizing process leads to a significant increase in the brittleness of the coating at a temperature below 750°C [12,22].”
- In the photo presented in fig. 10, 11, 12, 13, 22 increases are not marked or are not visible. This impairs the informativeness of the pictures.
As suggested by the Reviewers, the photographs of the hardness test sites from Fig. 11 to Fig. 14 have been removed and the average HV values have been presented in a table summarizing all samples for comparison.
Coating |
No. of the blade in the turbine palisade |
Coating thickness - SEM measurements |
|
Medium |
|
cross-section |
longitudinal section |
||||
APS
|
11 |
44 |
62 |
509 |
478 |
37 |
75 |
70 |
494 |
||
65 |
73 |
66 |
483 |
||
71 |
73 |
76 |
503 |
||
76 |
87 |
78 |
385 |
||
83 |
98 |
90 |
492 |
||
APS+VPA |
3 |
144 |
128 |
- |
530 |
5 |
132 |
132 |
502 |
||
30 |
141 |
138 |
523 |
||
32 |
128 |
126 |
543 |
||
75 |
134 |
128 |
504 |
||
84 |
146 |
138 |
595 |
||
APS+ |
13 |
locally |
locally do 92 |
453 |
424 |
27 |
76 |
108 |
443 |
||
55 |
locally 35-88 |
locally do 131 |
383 |
||
57 |
84 |
112 |
402 |
||
61 |
do 68 |
do 74 |
414 |
||
79 |
locally 94 |
locally 122 |
449 |
||
APS+TBC |
15 |
143 |
138 |
492 |
497 |
17 |
locally 89 - 96 |
locally do 142 |
449 |
||
21 |
144 |
148 |
498 |
||
7 |
134 |
128 |
510 |
||
43 |
128 |
119 |
502 |
||
45 |
136 |
124 |
523 |

Reviewer 2 Report
The study is quite interesting, but there are a number of questions and comments:
1) The Introduction does not fully describe the heat-resistant properties of the selected coating compositions (NiCoCrAlY, Ni3Al, ZrO2-Y2O3). It is necessary to give data on their heat resistance, based on high-temperature tests carried out in the literature. In addition, 24 references in the list of references is not enough for publication in the journal “Coatings”.
2) Table 1. Coating powders composition (Amdry 386-4 and MetCo204 BN-S) must be added.
3) Fig. 1 should be moved to the "Results" section.
4) Fig. 2. It will be visually plotted on the turbine operation diagram the temperature regime in which it is operated. According to this cycle, tests were carried out, after which the coatings are shown in Figs. 14–21?
5) Figs. 1(a, b). The porosity of the outer layer of the resulting coating is very noticeable. How do you rate it? Perhaps this is the reason for the premature destruction of these coatings.
6) Fig. 5. How do you interpret the obtained results? Are only APS+TBC coating promising?
7) Figs. 1, 14–21. Elements mapping is required.
8) For the convenience of the reader, Figs. 14, 16, 18, 20 combined into one. Do the same for Figs. 15, 17, 19, 21. Figs. 22-24 should also be combined into one. You will get a visual comparison for all four types of coating.
9) Lines 432-443. There is a description of the microstructure, but no figure.
Author Response
1) The Introduction does not fully describe the heat-resistant properties of the selected coating compositions (NiCoCrAlY, Ni3Al, ZrO2-Y2O3). It is necessary to give data on their heat resistance, based on high-temperature tests carried out in the literature. In addition, 24 references in the list of references is not enough for publication in the journal “Coatings”.
In the introduction, information on the aluminide and MCrAlY coatings was supplemented with the following provisions:
NiCoCrAlY overlay coatings, unlike diffusion coatings, provide greater independence from the substrate alloy, but also greater flexibility in design, because the compositions can be modified depending on the expected degradation mechanisms. Typical bond coatings contain at least 4 basic elements plus yttrium (Y), hafnium (Hf), tantalum (Ta). The presence of a significant amount of Cr gives these coatings excellent corrosion resistance combined with good oxidation resistance. NiCoCrAlY coatings typically exhibit a two-phase β+γ microstructure. The presence of γ increases the ductility of the coating, thereby improving the resistance to thermal fatigue. Ni and Co added together results in better corrosion resistance. Cr provides resistance to hot corrosion, but the amount that can be added is limited by the effect it should have on the substrate and the formation of Cr-rich phases in the coating. The Al content is usually about 10-12% by volume. NiCoCrAlY also usually contain 1% vol. by weight yttrium (Y), which improves the adhesion of the oxide layer, combines with the sulfur and prevents it from segregating into the oxide layer, which would otherwise damage its adhesion. Hafnium (Hf) additions play a similar role to yttrium. The addition of rhenium (Re) in MCrAlY improves isothermal or cyclic oxidation resistance and thermal cycle fatigue, and tantalum (Ta) increases oxidation resistance.
And
Heat resistance tests carried out by Kukla et al. [27, 28] showed that the oxidized aluminum coating was characterized by excellent durability and tightness of the protective scale, as no scale chips were observed on the surface of the tested coating. They found that the phase structure of the scale consisted mainly of NiAl and NiAl2O4 intermetallic phases and a stable α-Al2O3 oxide, which improves the corrosion resistance of nickel alloys. They found a significant improvement in the strength response of the MAR 247 nickel alloy by about 200 MPa during cyclic loading at 900°C. They confirmed that the use of an aluminide layer can effectively protect the parent material against such processes as oxidation, corrosion and wear.
And
In order for the ceramic coating to have a chance not to spatter during the first thermal cycle, it is important that its thermal expansion is close to the thermal expansion of the substrate. For a coating to be useful, it must also have very low thermal conductivity. Zirconium oxide (ZrO2) stabilized with yttrium oxide (Y2O3) is widely used for this purpose. The addition of 5-15% yttrium oxide stabilizes the zirconia in its high-temperature crystalline form, thus avoiding a phase transition in the operating temperature range. Zirconium oxide (ZrO2) ceramic meets both requirements, with a coefficient of thermal expansion of 11-13x10-6 K-1 and a thermal conductivity of approximately 2,3 W/(mK) at 1000°C for a fully compacted material, it can be additionally reduced by introducing porosity.
The list of literature has been increased by the following items:
- Kwolek, P.S.; Kolek, Ł.; Zagula-Yavorska, M.; Morgiel, J.A.; Romanowska, J. Effect of Pd and Hf co-doping of aluminide coatings on pure nickel and CMSX-4 nickel superalloy. Archives of Civil and Mechanical Engineering 2018, Volume 18 (4), pp. 1421-1429.
- Goral, M.; Pytel, M.; Ochal, K.; Drajewicz, M.; Kubaszek, T.; Simka, W.; Nieuzyla, L. Microstructure of Aluminide Coatings Modified by Pt. Pd, Zr and Hf Formed in Low-Activity CVD. Process. Coatings 2021, Volume 11, 421. https://doi.org/10.3390/coatings11040421
- Lin, Y.; Duan, F.; Pan, J.; Zhang, C.; Chen, Q.; Lu, J.; Liu, L. On the adhesion performance of gradient-structured Ni–P metallic coatings. Materials Science and Engineering: A 2022, Volume 844, 143170. https://doi.org/10.1016/j.msea.2022.143170.
- Kukla D.; Kopeć M.; Kowalewski Z.L.; Politis D.J.; Jóźwiak S.; Senderowski C. Thermal barrier stability and wear behavior of CVD deposited aluminide coatings for MAR 247 nickel superalloy. Materials 2020, Vol.13 (17), pp.3863-1-11. DOI: 10.3390/ma13173863.
- Kopeć M.; Kukla D.; Yuan X.; Rejmer W.; Kowalewski Z.L.; Senderowski C. Aluminide thermal barrier coating for high temperaturę performance of MAR 247 nickel based superalloy. Coatings 2021, Vol.11 (1), pp.48-1-12. DOI: 10.3390/coatings11010048.
The above literature presents research on the production of protective coatings with a gradient structure, especially with the use of palladium and hafnium coatings in the oxidizing zone, and tests on the heat resistance of oxidized aluminum coatings.
2) Table Coating powders composition (Amdry 386-4 and MetCo204 BN-S) must be added.
The composition of the Amdry 386-4 coating materials is shown in Table 2.
Table 2. Chemical composition of Amdry 386-4 coating material
Chemical composition of Amdry 386-4 coating material, % by weight |
||||||
Ni |
Cr |
Co |
Al |
Hf |
Y |
Si |
47,8 |
16,4 |
21,9 |
12,1 |
0,5 |
0,6 |
0,4 |
In Chapter 2, the following was added: ZrO2-8Y2O3 ceramic powder (Metco 204 BN-S) with a chemical composition of ZrO2+ 8% by weight. Y2O3 with a granulation range of -125+11μm. Due to the large size of the powder particles, its dimensions were reduced with an average particle size of dV50=4.5 μm. To reduce the size of powder particles, the ball milling process was used, where a single cartridge consisted of: (i) powder, (ii) zirconia balls, (iii) a dispersant preventing the formation of agglomerates, and (iv) ethanol as a cooling medium.
3) Fig 1 should be moved to the "Results" section.
Fig. 1 has been moved to the "Results" chapter.
4) Fig 2. It will be visually plotted on the turbine operation diagram the temperature regime in which it is operated. According to this cycle, tests were carried out, after which the coatings are shown in Figs. 14–21?
Operation in the ZMG idling speed range is a temperature in the range of 700 - 750°C. Operation in the ZMAX maximum rotational speed range is a temperature in the range of 1100 - 1200°C. Operation in the range of maximum afterburning ZMAX DOP is a temperature above 1350°C. According to the cycle in Fig. 2, engine tests were carried out in dynamometer conditions. The structures of the protective coatings shown in Fig. 14-21 come from the blades subjected to the engine test (during 52 hours) under dynamometer conditions.
5) Figs. 1(a, b). The porosity of the outer layer of the resulting coating is very noticeable. How do you rate it? Perhaps this is the reason for the premature destruction of these coatings.
The porosity of the outer layer of the protective coating is probably due to the fact that the coating material was not uniform in size and had an irregular shape with a large number of sharp edges. This is a consequence of their crushing and then grinding in ball mills. This form of the starting material may to some extent reduce the efficiency of the powder dosing process to the plasma stream, which has a significant impact on the properties of the coating, e.g. porosity, tightness. These observations result only from the analysis of the subject matter, and not from the research conducted on the grains (grain shape) of the purchased coating materials. Also, the presence of a discontinuity zone in the areas of the coating affects its porosity. It is estimated that the resulting coating has a porosity of up to 12%. Of course, this is an imperfection of the obtained structure affecting its durability. However, on the outer layer,
a network of cracks was not observed, which would cause delamination of the coating. It should be mentioned here that these coatings were made by experimental production (prototypes). In order to eliminate this type of imperfection of protective coatings, further technological research should be carried out to optimize the technical conditions for applying coatings, improving their adhesion, using numerical models and modern laboratory research methods.
6) Fig 5. How do you interpret the obtained results? Are only APS+TBC coating promising?
The protective coatings made using the APS+VPA and APS+TBC methods were not exhausted and retained their protective properties. Heat-resistant β-NiAl phase was preserved in both types of protective coatings. Thus, these coatings retained their heat resistance.
7) Fig 1, 14–21. Elements mapping is required.
The mapping has a wide range of descriptions and designations in the drawings. What details to mark on the microstructures in the drawings. The description is in the text.
8) For the convenience of the reader, Figs. 14, 16, 18, 20 combined into one. Do the same for Figs. 15, 17, 19, 21. Figs. 22-24 should also be combined into one. You will get a visual comparison for all four types of coating.
According to the reviewers' suggestion, the microstructure photographs Fig. 14, 16, 18, 20 were combined into one drawing. Also photographs of the microstructure Fig. 15, 17, 19, 21 have been combined into one drawing.
a) |
|
b) |
|
c) |
|
d) |
Fig. The structure of the protective coating in the blade cross-section A-A (see Fig. 3) made by the method: a) APS; b) APS+SLURRY; c) APS+VPA; APS+TBC (source: own)
a) |
|
b) |
|
c) |
|
d) |
Fig. The structure of the protective coating in the longitudinal section of the C-C blade (see Fig. 3) made by the method: a) APS; b) APS+SLURRY; c) APS+VPA; APS+TBC (source: own)
9) Lines 432-443. There is a description of the microstructure, but no figure.
The drawing is visible on the uploaded version.

Reviewer 3 Report
Design of heat-resistant protective coatings for turbine blades is important for decreasing the cost of operating the aircraft and increasing the safety of flights. This manuscript evaluated the heat resistance of prototype two-layer protective coatings applied to turbine blades, and found that an inner layer of the MCrAlY type applied to the blade by plasma spraying and an outer layer aluminized by diffusion by non-contact gas method protects the blade material against oxidation and ensures its thermal insulation. It's an interesting and meaningful work. But the structure of this manuscript and some figures need optimize. Therefore, the manuscript can be recommended for publication as a research article after conducting the following revisions:
1.Why the maximum thickness of the protective coating applied to the turbine blade is 160μm (APS)? Please give the necessary instructions.
2. Please give the scales in Figure 1 and analyze the thickness of the coating obtained by the different methods. Please adjust the same figure size for all the figures in this manuscript.
3. The scales in Figs 10-13 are not clearly shown and HV figures may be unnecessary. The average HV values are recommended to be provided directly by a table concluding all samples for comparison.
4. Figs 22-24 should have the same scale for comparison.
5. In order to make the paper readable, I recommend the author combine the results and discussion in the revision version. For example, after give the data of thermal diffusivity in Fig 5 and then discuss the research finding immediately.
6. Conclusions need be more concise and excessive discussions are recommended to be canceled.
7. What is the analysis method and the corresponding results for the chemical composition of the coating in Section 4?
8. Check grammar mistakes through the manuscript.
Author Response
1.Why the maximum thickness of the protective coating applied to the turbine blade is 160μm (APS)? Please give the necessary instructions.
When designing protective coatings, it was assumed that their tests would be carried out during a 50-hour engine test under dynamometer conditions. The authors had the RD-33 turbine engine at their disposal. Therefore, protective coatings have been applied to the turbine blades of this engine. The technical conditions for the blades of the RD-33 engine impose a maximum thickness of the protective coating of 160μm.
- Please give the scales in Figure 1 and analyze the thickness of the coating obtained by the different methods. Please adjust the same figure size for all the figures in this manuscript.
In Fig. 1, the scales of the coating microstructures have been introduced. The authors did not present the thickness of the coatings obtained by various methods in this article because this information was presented in another publication of the authors included in the bibliography under item 21, i.e. Ułanowicz L., Dudziński A., Szczepaniak P., Nowakowski M., Applying protective coating on the turbine engine turbine blades by means of plasma spraying. Journal of KONBIN 2020, Volume 50(1), pp. 193-213. Macro and microstructure studies of the protective coatings after they were applied to the turbine blades as well as after the engine test in the conditions of the aircraft dynamometer were carried out in various research centers. Therefore, the authors do not have the ability to adjust the size of the scales in all figures in this article.
- The scales in Figs 10-13 are not clearly shown and HV figures may be unnecessary. The average HV values are recommended to be provided directly by a table concluding all samples for comparison.
As suggested by the Reviewers, the photographs of the hardness test sites from Fig. 11 to Fig. 14 have been removed and the average HV values have been presented in a table summarizing all samples for comparison. Hardness distribution tests of APS, APS+VPA, APS+TBC protective coatings did not show significant differences in hardness on the leading edge of the turbine blade, so it can be concluded that the base material was not overheated. In the case of protective coatings of the APS+SLURRY type, the hardness distribution showed a difference on the leading edge of the turbine blade of up to 85 HV. This may indicate overheating of the base material.
Coating |
No. of the blade in the turbine palisade |
Coating thickness - SEM measurements |
|
Medium |
|
cross-section |
longitudinal section |
||||
APS
|
11 |
44 |
62 |
509 |
478 |
37 |
75 |
70 |
494 |
||
65 |
73 |
66 |
483 |
||
71 |
73 |
76 |
503 |
||
76 |
87 |
78 |
385 |
||
83 |
98 |
90 |
492 |
||
APS+VPA |
3 |
144 |
128 |
- |
530 |
5 |
132 |
132 |
502 |
||
30 |
141 |
138 |
523 |
||
32 |
128 |
126 |
543 |
||
75 |
134 |
128 |
504 |
||
84 |
146 |
138 |
595 |
||
APS+ |
13 |
locally |
locally do 92 |
453 |
424 |
27 |
76 |
108 |
443 |
||
55 |
locally 35-88 |
locally do 131 |
383 |
||
57 |
84 |
112 |
402 |
||
61 |
do 68 |
do 74 |
414 |
||
79 |
locally 94 |
locally 122 |
449 |
||
APS+TBC |
15 |
143 |
138 |
492 |
497 |
17 |
locally 89 - 96 |
locally do 142 |
449 |
||
21 |
144 |
148 |
498 |
||
7 |
134 |
128 |
510 |
||
43 |
128 |
119 |
502 |
||
45 |
136 |
124 |
523 |
Average blade material hardness: 405 – 420 HV.
- Figs 22-24 should have the same scale for comparison.
Macro and microstructure studies of the protective coatings after their application on the turbine blades as well as after the engine test in the conditions of an aircraft dynamometer were carried out in various research centers on various electron microscopes equipped with various systems, or metallographic microscopes. Therefore, the authors do not have the ability to adjust the size of the scales in all figures in this article.
- In order to make the paper readable, I recommend the author combine the results and discussion in the revision version. For example, after give the data of thermal diffusivity in Fig 5 and then discuss the research finding immediately.
The article was corrected according to the reviewer's suggestion.In Chapter 3, subsections (3.1, 3.2, …) have been deleted and Chapter 3 is a uniform text on the tests in question.
- Conclusions need be more concise and excessive discussions are recommended to be canceled.
Proposals have been redrafted in order to organize them. Removed e.g. sentences:
“In these coatings there is a phase of the protective oxide Al2O3, and also Ni3Al compounds and
a mixture of oxides. These coatings retained their heat resistance. It should be added here that the disappearance of the β-NiAl phase from the coating and the reduction of the aluminum content in it below 8% by weight eliminates such a protective coating from further work. In the case of the protective coating made with the APS+SLURRY method, it can be stated that too high activity of the Al source in the SLURRY aluminizing process leads to a significant increase in the brittleness of the coating at a temperature below 750°C [12,22].”
- What is the analysis method and the corresponding results for the chemical composition of the coating in Section 4?
Chemical composition tests (qualitative analysis) were carried out on scanning electron microscopes TM 1000, Hitachi SU 8000 and Quanta 3D FEG. TM 1000, Hitachi SU 8000 microscopes are equipped with an attachment for analyzing the chemical composition in micro-areas (EDS-Energy Dispersive Spectroscopy) by Thermo Noran. The tests were carried out on the cross-sections of the layers at the voltage accelerating the excitation electron beam V=15kV. The Quanta 3D FEG microscope is equipped, among others, with an integrated EDS/WDS/EBSD system (EDS X-ray energy dispersion spectrometer, WDS X-ray wavelength dispersion spectrometer and a backscattered electron diffraction analysis system EBSD). For the study of the distribution of elements on the cross-section of the leading edge of the blade, they were cut on an electro-spark saw at a distance of approx. 2 cm from the end of the blade blade, which allowed to obtain a cross-section of layers, and then embedded in epoxy resin with mineral filler using an ATM press. The samples prepared in this way were ground on abrasive papers with a gradation of 180 to 1200 and polished on a polisher with a disc saturated with diamond paste. Comprehensive structural and chemical composition tests for all prepared samples were carried out in the areas of the metallographic microsection, which are marked in the figure below. The analysis of the chemical composition of the coatings or individual interlayers was performed in the largest areas covering the observed area.
KN - around the leading edge in a perpendicular section at a distance of 15 mm from the tip of the blade, G - around the ridge in a perpendicular section at a distance of 15 mm from the tip of the blade, K - around the ridge in a perpendicular section at a distance of 15 mm from the tip of the blade, KS - around the ridge in a perpendicular section at a distance of 15 mm from the tip of the blade, Z – leading edge area in a perpendicular section at the base of the blade blade.
- Check grammar mistakes through the manuscript.
The article was translated into English by the Kwaśniewski Translators Team (www.ztk.pl).

Round 2
Reviewer 1 Report
The authors worked diligently to improve the manuscript of the article. Appropriate additions have been made. The article may be published in a revised version.
Reviewer 2 Report
The manuscript can be accepted for publication.